# The Impact of Social Support on Burnout among Lecturers: A Systematic Literature Review

**DOI:** 10.3390/bs14080727

**Published:** 2024-08-21

**Authors:** Beibei Cao, Norlizah Che Hassan, Muhd Khaizer Omar

**Affiliations:** Faculty of Educational Studies, Universiti Putra Malaysia, Serdang 43400, Selangor, Malaysia; gs62619@student.upm.edu.my (B.C.); khaizer@upm.edu.my (M.K.O.)

**Keywords:** social support, work stress, burnout, university lecturers, review

## Abstract

Background: Interest and concern regarding the social support and burnout of college lecturers have grown over the past decades. Maintaining good mental health is critical for university lecturers. Social support has been identified as an effective resource against burnout. However, few studies have comprehensively examined the connection between social support and burnout specifically among college lecturers. Therefore, this review aims to explore how social support influences the burnout of college lecturers. Methods: This study employed the systematic literature review (SLR) methodology. Results: A thorough systematic review of 20 studies was conducted between 2015 and 2024, drawn from five major databases: Web of Science, Scopus, APA PsycINFO, PubMed, and Eric. The review indicates that burnout is measured through the Maslach Burnout Inventory, Maslach Burnout Inventory—Educators Survey (MBI-ES), Maslach Burnout Inventory—General Survey (MBI-GS), Burnout Syndrome Evaluation Questionnaire, Copenhagen Burnout Inventory (CBI), Shirom–Melamed Burnout Measure (SMBM), and the Oldenburg Burnout Inventory (OLBI). While social support is measured through the Multidimensional Scale of Perceived Social Support (MSPSS) and Perceived Organizational Support (SPOS) survey. Individual and occupational factors contribute to lecturers’ burnout. This study reveals the association between social support and burnout among lecturers and it emphasizes the multifaceted role of social support in alleviating burnout among lecturers. Conclusions: The findings suggest that educational institutions should strengthen support systems and increase the emotional support available among lecturers to relieve burnout.

## 1. Introduction

Teachers are among the professionals who experience the highest levels of work stress and burnout, so an abundance of international research has recently focused on the burnout experience of teachers [1,2]. Burnout is thought to be caused by prolonged professional stress, particularly in human service professions like teaching [3]. College teachers are facing increased levels pressure and greater anticipation from their universities, students, and societies. Consequently, while the majority of teachers manage it well, some may feel stressed and burned out at work. When compared to their counterparts in 35 other countries, teachers in China indicated the second highest levels of emotional exhaustion [4]. Burnout among university instructors can be seriously affected by a variety of factors. When high levels of burnout are experienced by university lecturers, the results can include negative physical health effects (e.g., headaches and sleeplessness), mental health issues (e.g., sadness and anxiety), decreased job performance, reduced job satisfaction, increased turnover rates, and declining quality of education [5,6,7,8]. The phenomenon of burnout among teachers at universities is not only an individual psychological issue but a widespread sociological one with scaling effects. Therefore, maintaining good mental health is critical for university lecturers. The importance of social support has been extensively studied in relation to work-related stress and burnout experienced by university lecturers. Numerous studies have highlighted the positive effect of social support on the burnout of university lecturers [8]. High perceived levels of social support have been linked to reduced work stress and burnout, increased levels of mental health, and improved job performance [9]. The existing literature suggests that social support can contribute to managing burnout by increasing the social support available, and reducing the work-related pressure of lecturers should be a useful strategy for preventing teacher burnout.

### 1.1. Related Literature and Research Perspective

#### 1.1.1. Burnout

Researchers have studied burnout in a variety of professions, including educators, lawyers, social workers, and professionals in the medical and mental health fields [10]. Freudenberger et al. [11] provided the first definition of burnout in an article titled “Staff Burnout”, stating that it is typified by feelings of exhaustion and failure brought on by an overwhelming number of demands placed on an employee’s time, finances, resources, or spiritual strength. Maslach and Jackson [12] defined burnout as a syndrome that affects human service workers who have a lot of interpersonal interactions and is typified by emotional exhaustion, depersonalization, and reduced personal accomplishment. As conceptualized, the term “emotional exhaustion” refers to feelings of depression, anger, and frustration. A dehumanized and impersonal perspective on others that treats people more like objects or things than like real people is known as depersonalization. Reduced personal accomplishment is described as a decline in one’s self-efficacy at work and a tendency to have a negative self-perception [12,13]. A combination of individual, institutional, and social factors can lead to teacher burnout. A tendency toward burnout may be explained by certain demographic factors, including age, gender, working experience [14]. Organizational factors, such as the work environment and culture, social dynamics, and professional assistance, provide clarity on the burnout process. Sabagh et al. [15] emphasize that burnout is greatly influenced by the unfavorable demands of the job (such as workload, task characteristics, and value conflict) and the absence of resources (such as control, rewards, and social support). It was discovered that personal traits like motivation and optimism as well as external stressors (such as a lack of social support and family issues) result in burnout.

#### 1.1.2. Social Support

Social support has been found to be an effective resource for stress management [16] and people who received social assistance are resistant to the adverse consequences of stress; this definition emphasizes the effect of social support on stress. Social support is defined as the belief that one is respected, loved and cherished by others, and a part of a social network with obligations and mutual support [17]. According to Schwartzer et al. [18], social support is a multidimensional concept that refers to the nature and purpose of interpersonal connections people receive from others, including assistance, support, and care. In the context of research on burnout, this construct has been extensively studied. The materials highlight that social support falls into a number of categories. Social support can be categorized as instrumental support (e.g., pragmatic aid), informational support (e.g., advice, direction), emotional support (e.g., compassion, understanding), and companionship [19]. Apart from the nature of support, social support can also be distinguished based on its origin, which includes family, friends, and significant others [18]. In China, Xiao [20] pointed out three forms of social support: subjective support, objective support, and support utilization. According to the literature, friends in particular are regarded as a crucial component of happiness and as a significant source of both practical and emotional support.

Perceived social support refers to a person’s subjective assessment of the assistance they receive from their social surroundings. According to Zimet et al. [21], it describes the degree to which individuals think that their significant others, family members, and friends offer them instrumental, emotional, informational, and appraisal support when they need it. Regardless of the specific conceptualization, the reviewed literature emphasizes the crucial role of social support, in its various forms, for teachers’ work stress and burnout. Social support is identified as the resources offered by the social network that help individuals to deal with stress and burnout. This understanding of the different types, sources, and dimensions of social support is particularly relevant for university teachers. Investigating how these aspects of social support contribute to positive outcomes can provide guidance for creating interventions and support systems that work to enhance the overall mental health of teachers.

#### 1.1.3. The Relationship between Burnout and Social Support

Social support has long been understood as a successful strategy for managing burnout in the workplace. Research has consistently shown that the degree of burnout can be decreased by social support among teachers. In secondary education, according to Zhao and Chen [22], supervisor support is the type of social support that Chinese secondary school teachers receive the least amount of, despite it being the most successful in preventing stress, emotional exhaustion, and decreased accomplishment; social support can effectively mitigate teachers’ stress and burnout. Additionally, social support from the school principal was directly associated with less emotional exhaustion among Swiss primary and secondary school teachers [23]. In higher education, Xu et al. [24] learned about the level of burnout among university instructors and investigated the connection between burnout and social support, and found that there is a negative correlation between burnout and social support among university instructors. Studies have provided encouraging evidence for the influence of social support on burnout. Therefore, social support can directly reduce burnout among teachers [25,26,27].

Some researchers discovered that social support has produced a moderating effect: the relationship between stressful life events and internalized problem behaviors is moderated by social support [28]. Additionally, social support significantly reduces the relationship between work stress and occupational commitment. A large and growing body of literature has concluded that the relationship between work stress and burnout was partially mediated by social support in many professions. Duru and Balkis [29] came to the conclusion that social support plays a mediating role. Furthermore, social support acts as a mediator in the connection between cyberbullying and psychological health [30]. The relationship between work stress and burnout was sequentially mediated by perceived social support and job satisfaction among bank employees. Burnout was positively correlated with job stress but negatively correlated with perceived social support and job satisfaction [31]. Velando-Soriano et al. [32] explored the association between social support and burnout syndrome among nurses, and identified the risk factors of burnout among nurses in a systematic review. Further, Park and Shin [33] revealed that support from school personnel was significantly related to burnout, in a meta-analysis of special education teachers. Therefore, social support is a fundamental psychosocial factor that reduces burnout by providing the social support needed to cope with burnout.

### 1.2. Knowledge of Gaps and Aims of Review

Numerous studies conducted over the years have demonstrated a link between university teachers’ social support and burnout. The reviewed existing literature analyzes the impact of social support on burnout for nurses, primary and secondary school teachers, and special teachers. Despite these advancements, there remains a lack of thorough, comprehensive, and systematic reviews that collate the accomplishments of previous decades and pinpoint areas needing further research. In addition, there is a lack of systematic literature reviews on the relationship between social support and job burnout among university lecturers. Systematic literature reviews (SLRs) are essential for summarizing the state of knowledge on a particular topic, because they offer a broad overview of the evidence and can direct future research efforts [34,35]. Addressing the research gap, based on a review of the literature, the goal of this systematic review is to offer a comprehensive and current overview of empirical knowledge on how social support impacts the burnout of university lecturers, focusing on studies published in international, English-language, peer-reviewed journals from 2015 to 2024.

The guiding research question for this review is: What is the current state of empirical understanding of the connection between burnout and social support among college lecturers?

This primary question is supported by three specific sub-questions:

RQ1: What are the main research trends concerning social support and its relationship to burnout among university lecturers from 2015 to 2024?

RQ2: How are burnout and social support measured in these studies?

RQ3: What are the factors that affect burnout in these studies?

RQ4: How does social support influence the burnout of university lecturers, and what are the underlying mechanisms and moderating factors involved?

## 2. Materials and Methods

This study employed the systematic literature review (SLR) methodology which ensures that the research findings can be repeated by other researchers in the future. We performed a literature review to address the research question, which included going over and assessing preliminary information obtained from databases relevant to the study. The primary analytical method used in this study is a systematic literature review, which comprises searching and evaluating pertinent data from five databases. The study protocol was registered on the PROSEPRO database on July 2024 under the code: CRD42024561633.

### 2.1. Search Strategy

Web of Science (WOS), Scopus, APA PsycINFO, Eric, and PubMed are the electronic databases we use to search for the included articles, which were published from January 2015 to June 2024. The creation of the search strings was initially influenced by study parameters such as the research topic (social support and burnout, as well as the studies’ type (empirical) and participants (university lecturers). We observe that these parameters eventually evolved into official inclusion standards for the studies that were identified.

This review aims to summarize the impact of social support on university lecturer burnout. Various keywords were adopted: (1) burnout, burn-out, burned out; (2) social support, family support, informational support, emotional support, financial support, instrumental support; (3) lecturers, university instructors, college instructors. University lecturers, college instructors, and college lecturers may not be comparable across countries and regions. This research treats them as equal across these contexts, despite regional and institutional differences, since university lecturers generally share core responsibilities, such as delivering lectures, designing course content, and assessing student performance. And the primary role of lecturers, regardless of their title or rank, is to educate students. Treating lecturers as equivalent might be necessary to maintaining the scope and focus of the study. The Boolean operators AND or OR were used to match the keywords. After that, 1197 articles in total were mined from the five databases. The search strings were arranged in accordance with each database’s specifications with the subject librarian’s assistance, as indicated in Table 1.

### 2.2. Inclusion/Exclusion Criteria for Studies

There are two different categories of eligibility requirements—inclusion criteria and exclusion criteria—in this research. In order to find papers relevant to this study, the search engines SPIDER and PICO were primarily used in this systematic review. A comprehensive search can be efficiently retrieved using the PICO (population, intervention, comparison, outcome) and SPIDER (sample, phenomenon of interest, design, evaluation, research type) tools [36,37]. The present study used both tools to locate systematic reviews and meta-analyses by using exclusion and inclusion criteria. Table 2 contains a list of the inclusion and exclusion criteria that were applied in this investigation.

Papers were selected for consideration using the following standards: (1) publication dates span from 2015 to 2024; (2) the following keywords must be included: burnout, social support, and lecturers; (3) the article is written in English; (4) quantitative, qualitative, and mixed-methods studies were included to consider this research topic through different dimensions. The chosen period for this study is 2015–2024. This choice is the result of multiple considerations. Firstly, during this timeframe, online resources were widely used, which made it possible for us to examine the impact of social support on college lecturers. Secondly, concentrating on the years 2015–2024 guarantees access to current and relevant literature, allowing us to capture the latest trends and advancements in the understanding of this issue. Lastly, a narrower time frame allows for a thorough review of the literature, which improves our comprehension of college lecturer burnout.

Criteria for exclusion: (1) articles focusing on individuals outside the college lecturer population, such as primary and secondary school teachers or professors were excluded from the review; (2) articles that primarily focus on social support that is not related to burnout and studies not related to burnout or social support were excluded; (3) articles that do not directly address the research question were excluded; (4) non-empirical articles such as opinion pieces, editorials, theoretical papers, and literature reviews were subjected to the review process. Only empirical studies presenting original research findings were considered. (5) Articles in other languages were not reviewed because the focus of the analysis was on English-language publications. (6) Papers for which the full text is not accessible.

### 2.3. Data Extraction

It starts with the identification phase, where a comprehensive search across multiple databases yielded 1197 records: 459 from Web of Science, 6 from Scopus, 86 from PubMed, 636 from Eric, and 10 from APA PsycINFO. Before screening these records, duplicates (totaling 82 records) were carefully removed, thus narrowing the field to 1115 records for screening. A total of 256 reports could not be retrieved for detailed review due to reasons such as being reviews, conference papers, book chapters, and magazines (255 records) and the language used not being English (1 record). The next phase involved a rigorous screening process, focusing on the relevance of the titles, abstracts, and keywords to the research objectives. This led to the exclusion of a considerable number of records. Of the retrieved records, 839 articles in total were excluded from the review, with detailed reasons provided: 709 were not related to burnout, 120 were not related to college lecturers, 7 were not related to social support, 3 were review articles. Finally, 20 articles were included for further coding. The PRISMA flowchart (Figure 1) shows the number of articles included and excluded at each stage. Word and Excel by Microsoft Office were used for results and administration, the bibliography was edited and arranged using EndNote X9.

### 2.4. Quality Assessment of Included Studies

We used the Crowe Critical Appraisal Tool (CCAT), which is pertinent to this review since it can be used to evaluate a variety of study designs, such as mixed-method, quantitative, and qualitative studies, and to evaluate the research quality of each study that was included in the review. This is significant due to the variety of study designs employed in this review, CCAT is also extremely reliable. The Introduction, Background, Methods, Sampling, Data Collection, Ethical Matters, Results, and Discussion are the eight category items that make up the CCAT. There is a maximum aggregate score of 40 and a five-point rating system for each category item. The CCAT User Guide offers comprehensive guidelines and resources for scoring each category item. Two reviewers conducted the quality assessment, and disagreements were resolved by discussion and consensus. The study quality was divided into three categories: low quality (CCAT score < 25), medium quality (CCAT score 25–30), and high quality (CCAT score > 30). The CCAT places a strong emphasis on measuring and documenting results for every category rather than just the study’s overall score. By using this method, papers that perform well overall but poorly in one or more categories are kept from being less visible than those that perform well in every category. The CCAT scores and attributes for each of the 20 studies are presented in Table A1 in Appendix B (see Appendix A).

### 2.5. Synthesis and Analysis of Results

In the coding phase, we extracted and organized information in terms of burnout, social support, and lecturers. Each article was categorized by the authors, year of publication, country, title, research aims, research method, variables (independent variables, mediator variables, moderate variables, and dependent variables), participants, and results. Every article in every subcategory was given a descriptive code, which was then produced through inductive analysis. Theme analysis was the primary technique employed in the section on qualitative data analysis. It was used to summarize and classify the articles prior to creating the framework and ensuring that the data and framework were related. Themes that were taken from unprocessed data were coded, categorized, and refined using this method. The current study adheres to the six-step model proposed by Clarke and Braun. The following are the steps that were followed: (1) become acquainted with the data; (2) produce codes; (3) identify themes; (4) assess themes; (5) specify themes; and (6) elucidate themes [38,39].

We thoroughly examined the data and familiarized ourselves with the articles’ contents pertaining to the research question before starting the thematic analysis. In the next two steps, we first used an inductive approach to assign codes to the articles based on broad themes, which allowed themes to emerge from particular observations in the empirical studies. It is necessary to extract recurrent subject terms from the articles while concentrating on important elements like the author, publication year and nation, the goal of the study, the methodology, and the findings. First, we reviewed every established theme and eliminated any that had strong connections or overlaps. In the next step, we combined and clarified each of the shortlisted themes, and we kept modifying them until every sub-theme was incorporated into the primary theme. In the last stage, we further clarified and enhanced the themes with the first two authors coming to an agreement on each theme after discussion and taking the research question into account. A third author was consulted in case there were any disagreements. We identified and improved the themes following a review and validation of the original codes. The following section presents the collection of themes that were produced along with an analysis of them. We identified the following through an iterative process of classifying the initial codes into more general subjects and author discussions: (1) the various factors that influence burnout according to the reviewed research; (2) relationships between social support and burnout among lecturers.

## 3. Results

The results are shown in two categories: quantitative and qualitative. Question 1 will be addressed in the study’s qualitative section, while Question 2 will be addressed in its quantitative section. Even though the body of research on lecturer burnout is growing, there are as yet no review articles that offer a thorough analysis of how social support affects burnout among lecturers.

### 3.1. Contexts and Characteristics of the Studies

According to the findings of the word cloud analysis, burnout was mentioned the most, a total of 20 times, followed by teacher (14 mentions), education (10), job (7), and emotion (6) (see Figure 2). In recent years, there has been an increase in journal articles discussing the connection between social support and burnout. The 20 articles chosen for a systematic review from 2015 to 2024 are shown in Figure 3, which indicates the quantity of publications for each year. Starting with one publication in 2015, the output fluctuated in the following years with a minimal number of articles until a noticeable increase occurred in 2020 with four publications. From there, the trend shows an upward trajectory, particularly from 2020 onwards, signifying a growing interest in the topic. The chart peaks in 2023 with five publications, suggesting a significant rise in the research focus on how social support affects burnout among university lecturers. The early data for 2024 indicate two publications hinting at ongoing interest in the subject matter. Overall, Figure 2 reflects the academic community’s escalating attention on the effect of social support on burnout in higher education environments. Figure 4 presents a concise tabulation of the contributions of different countries to a body of research on the influence of social support on burnout among university lecturers. The figure indicates that the United States, Canada, and Iran lead in terms of research volume, with America contributing four publications, Iran four, and Canada three, underscoring their significant academic output in this domain, followed by Australia, Ethiopia, India, Mexico, Turkey, China, Portugal, Norway, and Saudi Arabia with one publication each. Most of the studies were found to have adopted a quantitative research design, while two articles adopted mixed methods.

### 3.2. Themes

This study investigates the complex connection between social support and burnout among college lecturers. The recent literature has increasingly recognized the effect of social support on lecturer burnout. Our review focuses on three core research questions: how burnout and social support are measured, the factors influencing burnout, and the specific mechanisms through which social support impacts the burnout of college lecturers.

#### 3.2.1. Measurement of Burnout and Social Support

Burnout is measured through seven different methods, including the Maslach Burnout Inventory (MBI), the Maslach Burnout Inventory—Educators Survey (MBI-ES), the Maslach Burnout Inventory—General Survey (MBI-GS), the Burnout Syndrome Evaluation Questionnaire, the Copenhagen Burnout Inventory (CBI), the Shirom–Melamed Burnout Measure (SMBM), and the Oldenburg Burnout Inventory (OLBI). Additionally, social support is measured through two distinct approaches—general social support and perceived social support.

Burnout is a complex construct assessed through various scales, each emphasizing different dimensions. The Maslach Burnout Inventory (MBI) and Maslach Burnout Inventory—Educators Survey (MBI-ES) have been used extensively in the literature to measure burnout. Five studies used the Maslach Burnout Inventory (MBI) to measure burnout [40,41,42,43,44,45]. The Maslach Burnout Inventory (MBI), which is the most widely used measure of burnout, is made up of three subscales: reduced personal accomplishment, depersonalization, and emotional exhaustion. Research shows the different levels of burnout of the lecturers based on a comparison of these dimensions. According to their findings, Akkaya and Serin [42] concluded that the personal accomplishment dimension is where academics experience burnout the most, followed by the depersonalization and emotional exhaustion dimensions. Another study demonstrated that most lecturers suffer from mild burnout, followed by extreme and low burnout [44]. In addition, six studies applied the Maslach Burnout Inventory—Educators Survey (MBI-ES) [2,46,47,48,49,50], which was created especially for educators and includes staff members, volunteers, administrators, and instructors who work in educational environments. In the Babb et al. [2] study, burnout measurement used the Maslach Burnout Inventory—Educators Survey (MBI-ES); the MBI-ES included 22 items that were broken down into three subscales: personal accomplishment (eight items), depersonalization (five items), and emotional exhaustion (nine items). Each subscale was scored on a seven-point Likert scale that went from one (never) to seven (every day). The results of the MBI-ES also show that lecturers have different degrees of burnout. Roohani and Dayeri [46] found teachers had a low degree of burnout. In another study, personal accomplishment was high among teachers, but emotional exhaustion and depersonalization were found to be low [47]. In the study by Kaiser et al. [51], the Maslach Burnout Inventory—General Survey (MBI-GS), whose scale goes from zero (never) to six (every day), was utilized to represent the fundamental aspect of burnout, and the results suggest that lecturers experienced lower levels of burnout.

In the study by Sánchez et al. [52], burnout was measured using the Burnout Syndrome Evaluation Questionnaire for Education Professionals (CESQT-PE). This is a 20-item instrument including four dimensions: illusion of work, psychic wear and tear, indolence, and guilt. A four-point Likert scale, with zero representing never and four representing every day, was used to rate the items [53]. And the results indicate that a significant proportion (30%) of academics were found to be at a low level of burnout, while the majority (33.6%) assessed their overall score on the CESQT-PE scale as being at a medium level [52]. Two studies from the literature applied the Copenhagen Burnout Inventory (CBI) [53,54]. This tool was devised by Kristensen et al. [55] and evaluates three types of burnout symptoms for people who work in the human services field: burnout related to work (seven items), burnout related to clients or students (six items), and general burnout (six items). Every item has a five-point rating system that goes from zero (never) to five (always), with average results for each subscale given. Scores ≥50 in each of the three subscales were deemed to represent high-level burnout. Each subscale’s score is the average of the item scores within the subscale and ranges from 0 to 100. In other research, 164 lecturers (49.5%) said they were highly burned out on a personal level, 121 lecturers (36.6%) revealed high levels of burnout from their jobs, and high levels of burnout related to students were reported by 36 lecturers (10.9%) [54].

In addition, Mota and Rad [56] used the Shirom–Melamed Burnout Measure (SMBM) to measure burnout, this 14-item instrument accesses three dimensions of burnout: physical fatigue (six items), cognitive weariness (five items), and emotional exhaustion (three items). It is also possible to compute an overall burnout. Participants responded on a seven-point Likert scale, with higher scores representing higher burnout levels, and with one representing never and seven representing always. The findings indicate that, with 24% of participants reporting high levels of overall burnout, Iranian teachers perceived themselves as experiencing high levels of burnout during the COVID-19 pandemic. Finally, burnout was measured with a six-point scale using items from the Oldenburg Burnout Inventory (OLBI) [57], which is a self-report measure used to assess the degree of burnout at work, and contains 16 items including two subscales measuring tiredness and disengagement. The results of this measurement show lecturers experienced moderate levels of burnout [9].

Social support is measured by social support scales or perceived social support scales.

Perceived social support was used by Taylor and Frechette [9] to gauge the amount of social support participants in their study had received. Zimet et al. [21] developed the Multidimensional Scale of Perceived Social Support (MSPSS) to gauge respondents’ perceptions of social assistance. The 12-item MSPSS is divided into three subscales: four items for family support, four for friend support, and four for significant other support. A seven-point Likert scale, ranging from one (very strongly disagree) to seven (very strongly agree), was employed by the MSPSS. The total score was 12–84, which indicated the degree of perceived social support. It is generally believed that a total score of 12–35 points represents a low level of perceived support, 36–60 points means medium perceived support, and 61–84 points means high perceived support; higher scores demonstrate greater degrees of perceived social support. In Li et al.’s [49] study, the Perceived Organizational Support (SPOS) survey was employed to measure perceived social support [58]. A five-point Likert scale, with one denoting “strongly disagree” and five denoting “strongly agree,” was used to rate the items. The scales in these studies do not analyze the degree of social support that university lecturers receive (Table 3).

#### 3.2.2. Factors Influence Lecturers’ Burnout

A combination of individual and organizational factors can lead to teacher burnout. Only three of the reviewed studies examined the connection between lecturer burnout and demographic traits. For instance, Mota and Rad [56] found that there were no significant differences between demographic variables except gender; men in Iran experience higher levels of burnout than women. While Castro et al. [54] concluded that female gender is linked to higher levels burnout. However, other research showed that burnout was not influenced by gender [9]. Studies have shown that locality has a significant effect on the burnout of educators, and rural teachers were more burned out than teachers who live in urban areas [44].

Researchers have primarily examined personal and occupational factors of burnout. Findings consistently show a relationship between personal factors and burnout. Padilla et al. [43] indicated that higher levels of burnout can be caused by time allocation and perceived pressure, lower levels of social support, family, sleep, and leisure time. In addition, Sánchez et al. [52] emphasized the connections between burnout and the assessed academics’ psychosocial factors, including issues with students, physical and mental effort, long workdays, and discontent with compensation. Kant and Shanker [44] found a negative correlation between emotional intelligence and burnout. Similarly, findings have revealed that the regulation of emotions and subjective well-being contribute to burnout [59]. Additionally, Castro et al. [54] concluded that burnout was positively related to reduced resilience and increased stress and depression, and changes in sleep patterns.

Research also indicates the occupational factors that have been identified as contributing to burnout. According to Roohani and Dayeri [46], burnout was linked to organizational factors, such as conflict, a lack of support from administrators, insecurity, demotivation, lack of autonomy, and student classroom behavior. Kaiser et al. [51] also proposed that burnout was linked to job demands, specifically workload, work–family conflicts, and work demands. Further, Li et al. [49] examined the connection between teaching–research conflict (TRC) and burnout, and discovered that TRC has a negative correlation with personal accomplishment but a positive correlation with emotional exhaustion and depersonalization. According to Akkaya and Serin [42], factors such as weekly course load, academic titles, and whether the participants have any administrative responsibilities all affect levels of burnout in higher education. Further, Samadi et al. [41] found a negative correlation between job satisfaction and burnout; they also found a direct positive correlation between intention to leave and burnout. Coyle et al. [60] found workload, control, reward, community, fairness, and values are identified as stressors impacting burnout. Beyond these areas, some additional factors also impacting burnout, including finances, staffing decisions, and external pressures. In addition, instructors have reported that their primary sources of burnout were work experience, level of education, nature of the position, organizational support, honor, and recognition [38]. Alamoudi [47] explored a negative relationship between work autonomy and burnout among EFL teachers; the teachers seem to have a high degree of perceived autonomy and a low degree of burnout at work. Therefore, research evidence demonstrates correlations between burnout and demographic, personal, and occupational factors. Fewer studies have examined the connection between demographic factors and burnout, and numerous studies have consistently shown occupational and personal factors affecting burnout among lecturers in the past ten years.

#### 3.2.3. Mechanisms That Impact Burnout through Social Support

The complex association between social support and burnout has been extensively studied, demonstrating the crucial role of reducing burnout in educational settings. A recent survey of the literature found that social support has a fundamental role in buffering burnout. Cormier et al. [61] revealed that supports like clear job expectations, autonomy, participation in decision-making, and coworker and manager support buffer burnout. We often experience higher levels of job contentment and engagement as well as a decline in burnout when we have the necessary and proper resources to complete our tasks. Babb et al. [2] found a negative correlation between administrative support and burnout. Administrative support promotes increased accomplishment and reduced exhaustion, and the higher burnout rates among lecturers would benefit from greater administrative support. Taylor and Frechette [9] considered that perceived social support has a negative correlation with burnout: it is a coping resource providing some assistance to relieve stress and burnout and it can reduce the high levels of burnout. Padilla and Thompson [43] support the moderator hypothesis, suggesting that reduced burnout is associated with increased social support, family time, leisure activities, and sleep. Social support at work in an academic environment can come from peers, departments, or colleagues. It is worth noting that acquiring workplace social support from the university and department can help reduce burnout. Perceived supervisor support (PSS), according to Li et al. [49], moderated the impact of teaching–research conflict (TRC) on depersonalization and tiredness, but it had no moderating influence on the connection between TRC and personal accomplishment. This study also found that psychological capital (PsyCap) moderated the influence of TRC on all the three aspects of burnout at work. The accumulated research strongly advocates that social support can relieve burnout among university lecturers. Support from friends, family, colleagues, and pets was helpful for lecturers in coping with work stress and burnout [62]. Previous studies have demonstrated the significant impact of social support on burnout among university lecturers. Specifically, most of the literature mentions support from coworkers, managers, family, friends, and pets. It can be seen from the literature that emotional support is the most effective in alleviating the job-related burnout of university lecturers, and emotional support was negatively related to burnout. This comprehensive understanding suggests that strengthening emotional support mechanisms can be a crucial strategy for reducing burnout and promoting mental health, particularly in educational settings.

## 4. Discussion

Burnout is measured through various scales, and numerous studies have explored personal and organizational factors contributing to burnout among university lecturers. Moreover, researchers have consistently investigated the significant effect of social support on burnout among lecturers. There are many measurements of burnout, including the Maslach Burnout Inventory, the Maslach Burnout Inventory—Educators Survey (MBI-ES), the Maslach Burnout Inventory—General Survey (MBI-GS), the Burnout Syndrome Evaluation Questionnaire (CESQT-PE), the Copenhagen Burnout Inventory (CBI), the Shirom–Melamed Burnout Measure (SMBM), and the Oldenburg Burnout Inventory (OLBI), while social support is evaluated by the Multidimensional Scale of Perceived Social Support (MSPSS) and the Perceived Organizational Support (SPOS) survey. There are many studies analyzing the degree of job burnout among university lecturers through various measurements, which have demonstrated that lecturers experience different degrees of job burnout. According to the MBI, most lecturers suffer from mild burnout, followed by extreme and low burnout. The results of the MBI-ES show that lecturers have a low degree of burnout, but the results also suggest that participants experienced low burnout measured using the MBI-GS. Additionally, the majority of lecturers experienced a medium level of burnout as measured by the CESQT-PE, and the majority of lecturers revealed high levels of burnout related to their personal lives and jobs, with only a small percentage of lecturers experiencing burnout associated with their students. According to the SMBM, lecturers perceive high levels of burnout, while the results of the OLBI show that lecturers experienced moderate levels of burnout. Therefore, the majority university lecturers experienced high or moderate levels of burnout, with a sense of personal accomplishment being high among university lecturers, followed by the depersonalization and emotional exhaustion dimensions. Regarding the social support scale, these studies did not analyze the degree of social support received by university lecturers; thus, future research should measure the degree of social support that university lecturers receive.

The literature suggests that personal and occupational factors result in teacher burnout. Among the teacher-level factors, an employee’s demographics may be able to explain a propensity toward burnout. Across the literature, a number of writers have commented that burnout is associated with gender as well as location, with male educators reporting feeling more burned out than female educators in Iran, and instructors in rural regions experiencing higher rates of burnout compared to their urban counterparts. Meanwhile, a recent survey of the literature found that higher levels of personal burnout are linked to the female gender. In addition, emotional intelligence and emotion regulation are negatively correlated to burnout. Recently, research found that higher levels of burnout are linked to lower levels of social support, family, sleep and leisure time, a lack of autonomy, and some personal traits like motivation and optimism. The most recent study on burnout has identified many occupational factors as having a significant impact on burnout, with the main domains identified being: workload, conflict, rewards, a lack of support, family and student stressors, and job satisfaction. Across the literature, social support is needed to reduce burnout, workplace stressors and the absence of social support are the major stressors causing burnout among college lecturers. In addition, some issues with students and family also contribute to burnout.

Numerous studies have explored the intricate connection between burnout, social support, job satisfaction and engagement, and teaching–research conflict, especially among university lecturers. Research consistently highlights the significant influence of social support. For instance, social support acts as a buffer against burnout, and different types of support and sources impact burnout. Social support is crucial in buffering burnout, and administrative support can reduce high levels of burnout among lecturers. Additionally, burnout was found to be correlated with perceived social support, which is a coping mechanism that helps lecturers alleviate high levels of stress and burnout. Furthermore, perceived supervisor support moderated the impact of teaching–research conflict on depersonalization and exhaustion, but it had no moderating effect on personal accomplishment. The support from colleagues, family, friends, and pets mentioned in the above studies all belong to the category of emotional support. Consequently, these studies highlight the importance of emotional support in reducing burnout among lecturers. These studies advocate for strengthening emotional support mechanisms as the most effective strategy for reducing burnout, particularly in educational settings.

This study makes several important contributions to our understanding of the influence of social support on burnout among college lecturers. First, it provides a comprehensive synthesis of the various measurement tools used to assess burnout; this research offers a robust framework for operationalizing and quantifying burnout across multiple dimensions. Moreover, the review underscores the critical role of perceived and general social support, measured through validated instruments. In addition, findings show that different personal traits and occupational stressors, both within and outside the workplace, result in burnout among lecturers, and this study can help instructors and university to recognize the antecedents of burnout and assist them in implementing corrective actions to deal with these issues and raise the standard and efficiency of education through eliminating the antecedents. It can change the status quo of university lecturer burnout through social support and provide indications as to how to assist lecturers in developing coping skills that will help reduce burnout related to their jobs. A key contribution lies in synthesizing extensive empirical evidence that illuminates the significant influence of social support systems on burnout. Investigating the connection between social support and burnout is of great significance for lecturers to reduce burnout, maintain physical and mental well-being, improve job satisfaction, reduced turnover intention, and promote the healthy and sustainable development of higher education.

The systematic literature review presented here has several limitations that could affect the breadth and depth of the results. Firstly, other significant types of literature like books, chapters, theoretical papers, and reviews were not included in the review because it was limited to empirical studies. Additionally, the focus was limited to peer-reviewed journal articles, thereby excluding recent conference proceedings and book chapters that might contain emerging ideas. Moreover, the review’s limitation to English-language publications may have resulted in the neglect of significant research published in other languages. This linguistic constraint raises the possibility that, in order to provide a more global perspective, future research may entail international collaborations that make it possible to access and incorporate literature in multiple languages. The use of descriptive and thematic analysis, rather than meta-analytical techniques, means the results are intended to explore themes and patterns rather than produce summaries of statistical findings. Although this method is useful for extracting in-depth stories and insights, it might not provide enough evidence to support conclusions about the efficacy of the interventions or other researched phenomena. More important, we need to explicitly acknowledge the limitation that our analysis did not differentiate between various forms of social support. This will help clarify that, while our findings offer valuable insights into the general role of social support, future research could benefit from a more granular examination of these effects.

Regardless of these drawbacks, this analysis offers guidance for future studies and contributes insightful knowledge to this quickly expanding and significant field in the current global setting. Cross-sectional designs were used in the majority of the reviewed studies, which makes it more difficult to determine causal relationships. Future study on the influence of social support on burnout should take into account longitudinal and experimental studies in order to overcome this limitation and provide more solid evidence. These kinds of studies would also make it possible to test interventions meant to improve social support and eliminate the antecedents of burnout. Additionally, it is also crucial to look into the intersectionality of different demographic and identity factors, such as gender, race, ethnicity, and sexual orientation, given the varying effects of social support sources and types across cultures. This would facilitate comprehension of how these elements influence social support experiences and burnout. Incorporating qualitative methods could provide rich insights into the lived experiences, perceptions, and meanings associated with social support and burnout among lecturers, complementing the primarily quantitative approaches used in the reviewed studies. Finally, building upon the theoretical and practical implications of previous studies, future studies should concentrate on creating, implementing, and carefully assessing interventions and programs meant to improve social support and reduce burnout among lecturers. Through addressing these research directions, researchers can enhance our comprehension of the complex connection between social support and burnout. This will help shape the development of more efficient and customized interventions and services for college lecturers.

## 5. Conclusions

The findings of this comprehensive review of research on social support and burnout among college lecturers offer several important theoretical implications. Firstly, the findings show that various personal and occupational factors both within and outside the workplace result in burnout, and the studies reinforce and extend existing theoretical frameworks that posit social support as a critical resource for coping with lecturer burnout. Moreover, the research illuminates the effect of social support on burnout, contributing to a more nuanced understanding of these relationships. Furthermore, the research clarified the differing effects of various types and sources of social support.

As outlined in the reviewed studies, individual-level interventions are highly advised to address burnout. Lecturers need to improve their skills in classroom management and adopt professional growth as instructors. These techniques could be used both after training and as part of their preparation for higher education; they also need to know their personality characteristics in order to reduce burnout at work. In this context, career counselors should evaluate lecturers’ personalities, and we think that institutional initiatives to prevent burnout are also necessary. The study emphasizes the importance of creating supportive social environments within educational settings. This can inform the development of evidence-based interventions and programs targeted at bolstering lecturers’ social support networks, both on and off campus. For example, universities could implement initiatives that facilitate peer support groups and opportunities for meaningful social connections among students. Additionally, efforts could be made to enhance family support systems and encourage positive relationships with significant others. Furthermore, universities should provide lecturers with more conducive work environments free from burnout, and they should also strongly motivate faculty members to collaborate in securing grants and providing services in order to foster peer support. Institutions can support departments and colleges by giving faculty administrative support in finding, writing, and submitting grant applications. College administration should examine college instructors’ job burnout from the higher education management point of view. One significant subset of instructors in colleges and universities are college lecturers. In spite of this, university administration does not give them the required attention. In order to determine the current levels of burnout and the primary stressors thereof, the administration departments of the universities should actively evaluate their job-related burnout. Moreover, lecturers themselves must gain a deeper comprehension of job burnout. Three dimensions of job burnout should receive more focus from university administration, since they provide the cornerstone of a successful burnout intervention or prevention.

## Figures and Tables

**Figure 1 behavsci-14-00727-f001:**
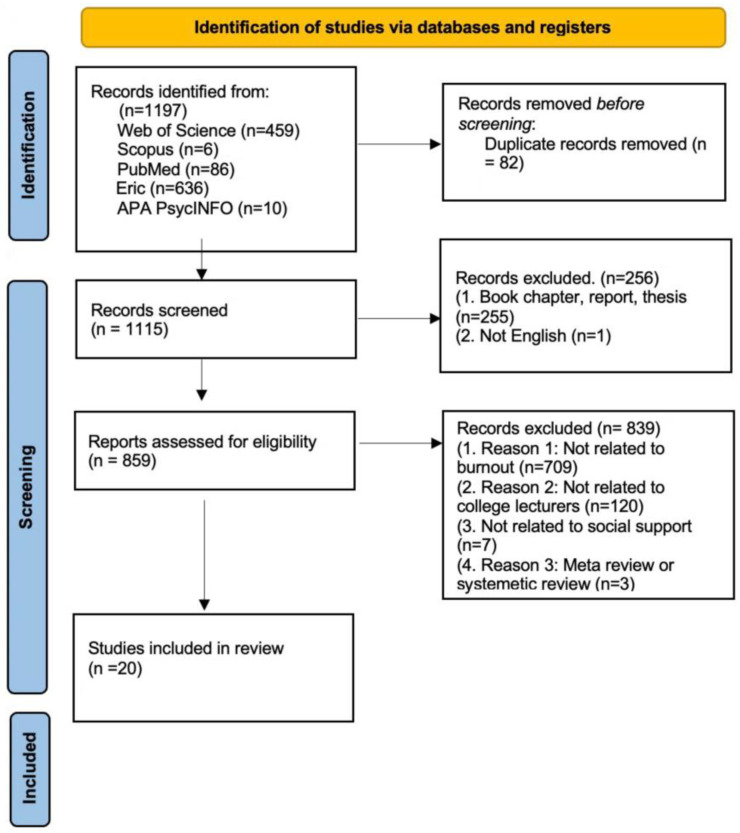
PRISMA flow diagram.

**Figure 2 behavsci-14-00727-f002:**
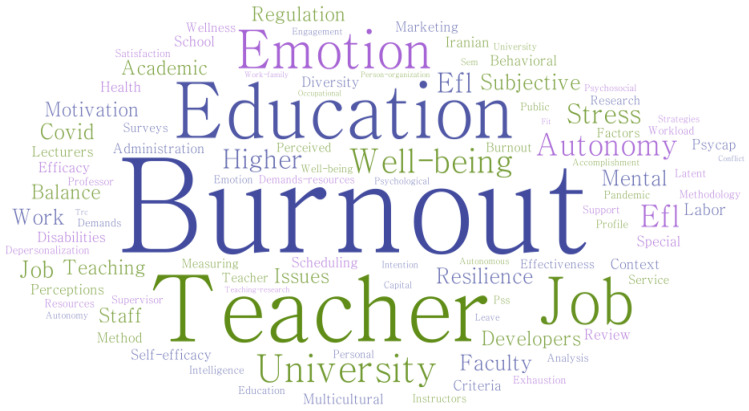
Word cloud map generated for 20 documents.

**Figure 3 behavsci-14-00727-f003:**
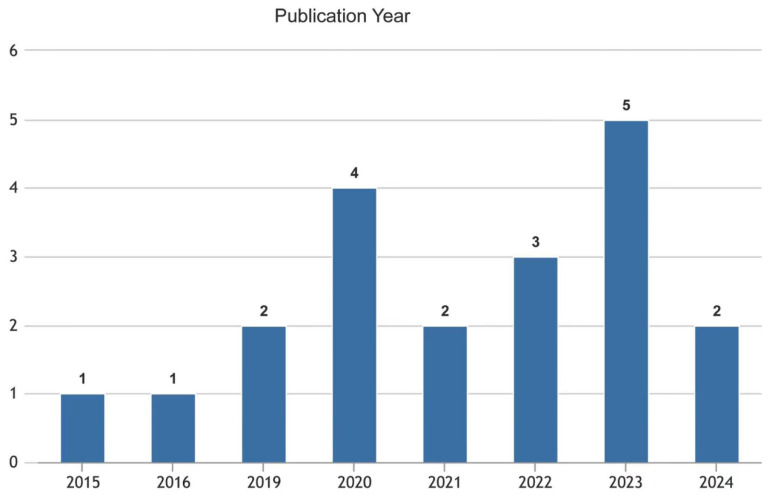
Publications arranged by publication year.

**Figure 4 behavsci-14-00727-f004:**
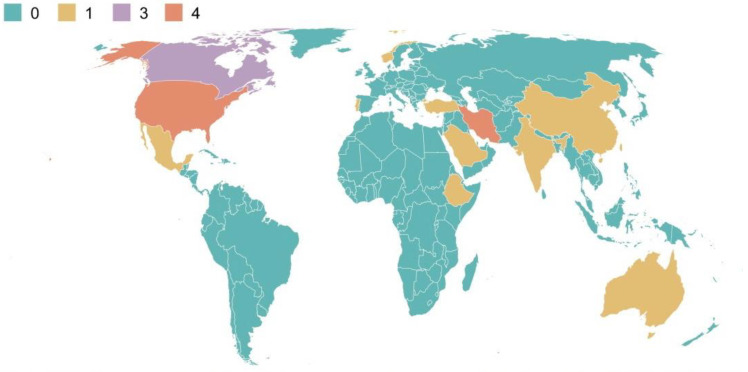
Geographical distribution of journal articles.

**Table 1 behavsci-14-00727-t001:** Search strings.

Search Query	Keywords in Title, Abstract, and Keyword
1	“burnout” OR “burn-out” OR “burned out”
2	“Social support” OR “family support” OR “informational support” OR “emotional support” OR “financial support” OR “instrumental support”
3	“lecturers” OR “university instructors” OR “college instructors”
Final	1 + 2 + 3

Databases: Web of Science, Scopus, APA PsycINFO, Eric, PubMed. Publication Dates: 1 January 2015 to 31 May 2024.

**Table 2 behavsci-14-00727-t002:** Inclusion and exclusion criteria.

Criteria	Inclusion	Exclusion
Population	College lecturers	Primary and secondary school teachers, professors
Interventions	None	None
Comparisons	None	None
Country setting	Any country	None
Outcomes	Studies that analyze relationships between social support and burnout	Social support not related to burnout
Setting	Studies with broad definitions of social support and burnout	Studies not related to social support or burnout
Language	English or translated into English.	Not in English
Study types and designs	Quantitative research Qualitative research Mixed Research	Studies without empirical data

**Table 3 behavsci-14-00727-t003:** Burnout and Social Support Measurement.

Variables	Measurement
Burnout	Maslach Burnout Inventory—Educators Survey (MBI-ES)
	Maslach Burnout Inventory—General Survey (MBI-GS)
	Burnout Syndrome Evaluation Questionnaire (CESQT-PE)
	Copenhagen Burnout Inventory (CBI)
	Shirom–Melamed Burnout Measure (SMBM)
	Oldenburg Burnout Inventory (OLBI)
Social support	Multidimensional Scale of Perceived Social Support (MSPSS)
	Perceived Organizational Support (SPOS)

## Data Availability

Not applicable.

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
