# Peer review of "The Impact of Social Support on Burnout among Lecturers: A Systematic Literature Review"

_behavsci, 2024, doi:10.3390/bs14080727_

Round 1
Reviewer 1 Report
Comments and Suggestions for Authors
The authors focused their systematic review on last nine years. The rationale behind selecting this specific timeframe need to be clearly described. In the introduction section, the latest review and meta-analysis should be highlighted, along with the specific gap that this review aims to fill.
It is recommended that the authors consider mediating factors involved between social support and burnout.
The exclusive focus on college lectures requires a more in-depth justification, as comparing college lectures with primary and secondary school teachers could provide valuable insights.
The table summarizing all identified studies should include the most relevant information, such as moderators, mediators, study type (longitudinal studies or cross-sectional). Both the review and the meta-analysis should be incorporated, with an indication of how many multilevel studies were conducted and their outcomes.
A detailed description of the the Crowe Critical Appraisal Tool (CCAT) should be included in the text.
The literature review should explore differences in results when comparing different measurements of burnout and social support.
A section on the underlying mechanisms between social support and burnout should be included, accompanied by a discussion of the findings.
In the discussion, it would be be beneficial to highlight the most important conclusions and practical implications derived from the considered studies, as well as any differences in results when using different instruments.
Reviewer 2 Report
Comments and Suggestions for Authors
The well-being of university educators is understudtied in the literature, so I think it is a meaningful work that provides a systematic review of the relationship between social support and lecturer's burnout, thereby enhancing our understanding about the status quo of the research field. I have to concerns regarding the study.
1) From the literature review, I find there are at least three ways to conceputalize social support. The first is to focus on its attributes or dimensions, such as emotional, financial, informational support. The second is about the sources of social support, like family and friends. The final is subjectivity vs objectivity of social support. Nevertheless, the authors only searched keywords like social support (in line 166 and 167, the authors mentioned twice, I guess there should be a typo here), family support, emotional support, informational support, financial support, instrumental support. I worry whether this list is comprehensive. Moreover, when they analyze the data, do they determintate the effects of social support and compare them between different conceptualization? I mean, for example, family support, let's say, may be good for lecturer because of its emotional rather than informational dimension. But, such analysis or discussion is limited in the manuscript.
2) This study focus on university lecturers, but something they refer it to instructors or college instructors. However, lecturers and instructors can be different concepts, and university and college can also be different types of institutions. In some countries, college does not gain university status or tend to be teaching oriented institution, while university is research oriented institutions. In some countries/regions (like US, UK, HK), lecturers are teaching staff (different from academic track like assistant, associate, and full professor), but some countries (like China, Australia) regard lecturers are academic staff equaltive to assistant professor. Moreover, to many countries and regions, there are no the post of instructors or the instructor position, (some may even treat it as teaching assistant) is ranked as lower than lecturers. In other words, college instructors, college lecturers, university instructors, university lecturers may not be comparable across countries and regions. Therefore, the authors need to justify why they treat them as equal across the contexts. Or they should use a broader concept to cover all of them, such as educators/teachers in higher education (it is not the good term, but I cannot think a better one this moment).
Round 2
Reviewer 1 Report
Comments and Suggestions for Authors
The authors have responded to my questions.